# A Bilevel Linear Programming Model for Developing a Subsidy Policy to Minimize the Environmental Impact of the Agricultural Sector

Konstantinos Ziliaskopoulos [1],* and Konstantinos Papalamprou [2]

1   Department of Environmental Sciences, University of Thessaly, 41500 Larissa, Greece
2   Department of Electrical and Computer Engineering, Aristotle University of Thessaloniki, 54124 Thessaloniki, Greece; papalamprou@ece.auth.gr
*   Correspondence: zilikons93@gmail.com; Tel.: +30-69-8703-0520

**Abstract:** The agro-food industry, while critical for establishing food security, is the most environmentally impactful industry since it causes biodiversity loss and the conversion of natural land to farms and pastures, requires pesticide and fertilizer use as well as high water consumption, and leads to greenhouse gas emissions as well as soil and environmental degradation. This impact can be mitigated through proper policy design. Environmental policy in agriculture, however, is inherently complex, due to the conflict between actors in the system, namely policy makers and farmers. This article introduces a bilevel linear programming (BLP) approach for the development of subsidy policies with the upper-level objective being the minimization of the environmental impact of the agricultural sector. Both levels of the model are formulated as linear programs and by considering the Water-Energy-Food-Climate Nexus, a general-purpose model is introduced. The methodology of the model formulation is spelled out. Finally, different approaches for fine tuning the BLP model are discussed in order to adjust it to each case study's needs, and the model is applied to the case study of the region of Thessaly, Greece.

**Keywords:** water-energy-food nexus; bilevel linear programming; agricultural sector; subsidy policy; environmental impact

## 1. Introduction

One of biggest global challenges is balancing agricultural production with environmental and ecological preservation. The food industry has far-reaching effects on all environmental and sustainability sectors, including water availability and quality, energy use and production, biodiversity, greenhouse gas emissions and global warming. These interlinkages have been a recent research focal point called the Water-Energy-Food-Climate Nexus [1], while the worldwide goals and key performance indicators are provided by the United Nation's 17 Sustainable Development Goals (SDGs) [2,3]. Visualizing, modeling, and optimizing the interactions between actors in the agricultural system, including farmers and policy-makers, can be done with a plethora of tools, such as heuristic algorithms [4], System Dynamic Models (SDMs) [5], empirical statistical models, etc. However, the application of multi-level optimization with operations research principles in the Nexus field of research has not been well explored in recent years. Multi-level linear programming models, while powerful tools to model interactive systems, are rarely used today due to their inherent complexity in their implementation and solution. The purpose of this article is to introduce a baseline bilevel linear programming (BLP) model that seeks to minimize the environmental impact of the agricultural sector through a subsidy policy.

An early attempt to apply BLP techniques to modeling government structures and decision making is discussed in [6], where the relationship between central government and government ministries represents the two levels of an optimization problem. Several

decades ago, Candler and Norton [7] applied BLP between the government and the private sector for an economic policy application where the government uses taxes and subsidies to achieve pre-determined goals, and the private sector reacts to those measures by forming an optimal plan of action. As a result, the authors showed that when individuals' objectives are recognized as part of the problem, the profit for the community could be doubled. In a recent study, Bostian et al. [8] use BL optimization in a different way, to include not only two levels of actors or institutions, but also the spatial aspects—agricultural nonpoint source pollution and spatial heterogeneity of both production and pollution—accounting for the interlinked nature of farm-level management decisions and their relation to agri-environmental policy incentives.

Anandalingam and Apprey [9] examined a method of conflict resolution by proposing adding a "referee" in so-called "Stackelberg games" [10]. They modelled such problems as linear and multilevel and suggested various solver algorithms. As an example, they presented a water dispute between India and Bangladesh, showing that both countries would benefit from a third party overseeing the deal, such as the United Nations.

The effect that incentives might have on the decisions that individual farmers take in terms of crop selection and farmland allocation, while considering price and yield risk has also been studied with different methodologies. Basnet et al. [11] employ a Bayesian econometrics risk-programming approach and a micro-economics analysis for the impact of decoupled payments on the agricultural sector in the European Union. Compared to BLP, such analysis is based on a statistical approach and includes risk assessment, while the former identifies and optimal solution while satisfying a series of constraints. In recent research, Barnhart et al. [12,13] applied a bilevel optimization model in two different case studies, one in Iowa, USA [12], and one in the Tully catchment in Australia [13], using policy maker and farmer interaction to both optimize water quality and minimize fertilizer use. Finally, Whittaker et al. [14] apply a hybrid of bilevel optimization and genetic algorithms to create a multi-objective optimization model with spatial components for Oregon, USA. This article aims to develop a mathematical modeling analysis of the interaction between government subsidy policy and individual farmer crop allocation and the associated environmental impact of agricultural practices. While individual vectors of the agricultural sector have been modeled and optimized through bilevel models and policies, this article introduces a baseline bilevel model with an interchangeable "environmental impact" parameter, designed to be easily adaptable to each case study's needs of optimization and allow interchangeability of ecological optimization parameters, including energy use, water quality and consumption, fertilizer use, pesticide use, greenhouse gas emissions and soil quality. To achieve this, the article provides a step-by-step approach to creating a BLP model for the agricultural sector, while it recommends solution methods, possible implementations, as well as improvements that can make it adaptable for a broad number of applications. The approach consists of two levels of optimization models: a lower-level model (LLM) aiming to capture the behavior of the individual farmer and an upper-level model (ULM) that models the policy-maker's strategy in issuing subsidies. The main goal of the model (at the upper level) is to minimize the environmental impact of the agricultural sector while anticipating the crop diversity employed by the individual farmers, as they are adjusting it to the issued subsidies to maximize their profits. The model is applied for the case study of the region of Thessaly, Greece, but it is setup in such a generic way that it can be modified by the user to minimize whichever environmental impact the case study requires, such as water depletion, greenhouse gas emissions, pesticide use and associated pollution, and others.

In this article, we present BLP fundamentals, the two levels of the model and the methodology used to transform the BLP model into a single level model in Section 2. The details of the case study are also described in Section 2.4. In Section 3, we present the unified baseline single level optimization model, as well as supplementary constraints used for the case study. Finally, discussion of the results and the relevance of this methodology in a WEF Nexus analysis are presented in Section 4. The limitations and applicability of this

methodology are also discussed. The article ends with conclusions and recommendations for future study.

## 2. Materials and Methods

Bilevel programming is similar to standard mathematical optimization [15], except that the constraint region is modified by including a defined linear objective function; it is a nested optimization model involving two problems, an upper one and a lower one [16]. Such models can be depicted as in Equations (1)–(6):

$$\min f_1(x, y*) \tag{1}$$

$$s.\ t\ g_1(x, y*) \leq 0 \tag{2}$$

$$h_1(x, y*) = 0 \tag{3}$$

$$y* \in argmin\ f_2(x, y) \tag{4}$$

$$s.\ t\ g_2(x, y) \leq 0 \tag{5}$$

$$h_2(x, y) = 0 \tag{6}$$

where the decision variables are split into two groups: the ULM variables $x$ and the LLM variables $y$. There are also two groups of constraints: Equations (2) and (3) are constraints of the ULM, and Equations (5) and (6) constraints of the LLM. The bilevel model is also hierarchical; the ULM contains the optimal solutions of the LLM, without necessarily the opposite being true. In the LLM, the decision variables of the ULM are parameters, not variables. If $f_1, f_2, g_1, g_2, h_1, h_2$ are linear functions, then the model in Equations (1)–(6) is a BLP model, which makes its solution considerably easier [17].

In the model constructed in this article, each level of optimization is modelled as a linear programming model, with the LLM for the individual farmer maximizing his profit and the ULM for the governmental body minimizing environmental impact. In order to simplify the resulting BLP model, this article uses a subsidy policy that fixes the subsidy amount per crop to a predefined level, while leaving the maximum subsidized farmland as a decision variable for the ULM.

### 2.1. The Lower Level Model

The LLM is constrained by the farmer's available farmland and the subsidies given by the ULM, while its objective function seeks to maximize the individual farmer's profit. Therefore, the LLM is modelled as such:

$$max\ \sum_{i \in S}(p_i - c_i)o_i x_i^k + \sum_{i \in S}(s_i + (p_i - c_i)o_i)X_i^k \tag{7}$$

$$s.t\ \sum_{i \in S}\left(x_i^k + X_i^k\right) \leq L^k \tag{8}$$

$$X_i^k \leq Y_i\ ,\ \forall i \in S \tag{9}$$

where:

**Sets**

   *S:* Set of available crops $i$
   *K:* Set of all farmers $k$

**Parameters**

   $p_i$: Profit of crop $i$ per unit of product
   $c_i$: Cost of crop $i$ per unit of product
   $o_i$: Yiel d of units of product of crop $i$ per unit of farmland
   $s_i$: Subsidy per unit of farmland with crop $i$

$L^k$: Available units of farmland of farmer $k$

$Y_i$: Maximum subsidized units of farmland per crop $i$

**Variables**

$x_i^k$: Units of farmland of non-subsidized crop $i$ of farmer $k$

$X_i^k$: Units of farmland of subsidized crop $i$ of farmer $k$.

Note that $Y_i$ is considered a parameter for the LLM but is derived by the ULM, where it is a decision variable, whereas $x_i^k$ and $X_i^k$ are decision variables for the LLM but parameters for the ULM.

*2.2. The Upper Level Model*

The ULM captures the decision-making process of the governmental body that issues the subsidy policy and seeks to minimize the environmental impact of the LLM in its objective function. As shown in Figure 1, the ULM seeks to optimize the agricultural environmental impact of the LLM. However, since the government cannot directly control the actions of the farmers, a multi objective optimization problem is required. The ULM is constrained by the total budget available for subsidies. Additional constraints can be added conditionally depending on each case study's needs. Therefore, the ULM is modelled as such:

$$min \sum_{i \in S, k \in K} e_i \left( x_i^k + X_i^k \right) \tag{10}$$

$$s.t \sum_{i \in S, k \in K} s_i X_i^k \leq B \tag{11}$$

where:

**Sets**

$S$ : Set of available crops $i$

$K$: Set of all farmers $k$

**Parameters**

$x_i^k$: Units of farmland of non-subsidized crop $i$ of farmer $k$

$X_i^k$: Units of farmland of subsidized crop $i$ of farmer $k$

$e_i$: Environmental impact of crop $i$ per unit of farmland

$s_i$: Subsidy per unit of farmland with crop $i$

$B$: Total available budget for subsidies

**Variables**

$Y_i$: Maximum subsidized units of farmland per crop $i$.

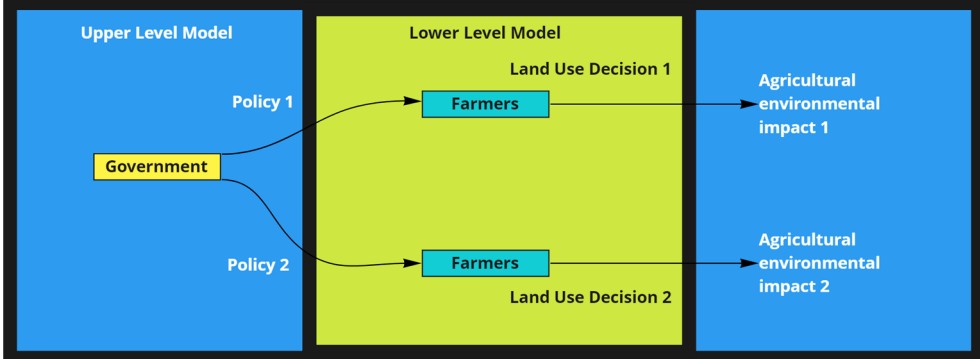

**Figure 1.** The interaction of the two levels of the model.

The parameter $e_i$ denotes the environmental impact of crops and can be replaced with the metric to be minimized in each case, such as water usage, energy usage, greenhouse

gas emissions, or pesticide use. Of course, due to the Water-Energy-Food-Climate Nexus, minimizing one resource's usage in the agricultural sector might positively influence all other interlinked resources. For example, minimizing the water usage for irrigation would also reduce the overall energy usage, since a majority of the energy used in agriculture is used for pumping out groundwater. Of course, more conditional constraints could be added to the ULM depending on each case's needs, as discussed later.

### 2.3. Formulation of Single Level Model

As shown in Figure 1, the ULM, the government, does not directly impact the environmental effects of the LLM. In order to optimize for the ULM directly while also predicting the response of the LLM, the multi-objective function problem is combined into a single-level optimization model. The two levels of the model are combined into an equivalent single level by employing the complementary slackness conditions (or Kuhn-Tucker conditions) of the LLM [18]. The complementary slackness conditions are added as constraints, denoting that the product of the primal decision variables with the corresponding dual constraints equal zero, while the product of the decision variables of the dual with the corresponding primal constraints equal to zero. According to the Kuhn-Tucker conditions, if a mathematical model's primal constraints, dual constraints, and complementary slackness conditions are all fulfilled, then the solution is the optimal solution for the model. This way it is possible to optimize the LLM without including its objective function, by adding the LLM's primal constraints, dual constraints, and complementary slackness conditions into the ULM, basically formulating the BLP model into a unified model. In Equations (12)–(17), the dual constraints and the complementary slackness conditions of the LLM are presented.

$$u^k \geq (p_i - c_i)o_i, \forall i \in S, \forall k \in K \tag{12}$$

$$v_i^k + u^k \geq s_i + (p_i - c_i)o_i, \forall i \in S, \forall k \in K \tag{13}$$

$$(\sum_{i \in S} \left( x_i^k + X_i^k \right) - L^k) * u^k = 0, \forall k \in K \tag{14}$$

$$(X_i^k - Y_i) * v_i^k = 0, \forall i \in S, \forall k \in K \tag{15}$$

$$\left( u^k - (p_i - c_i)o_i \right) * x_i^k = 0, \forall i \in S, \forall k \in K \tag{16}$$

$$\left( v_i^k + u^k - s_i - (p_i - c_i)o_i \right) * X_i^k = 0, \forall i \in S, \forall k \in K. \tag{17}$$

where:

**Sets**

> *S:* Set of available crops *i*
> *K:* Set of all farmers *k*

**Parameters**

> $p_i$: Profit of crop *i* per unit of product
> $c_i$: Cost of crop *i* per unit of product
> $o_i$: Yield of units of product of crop *i* per unit of farmland
> $s_i$: Subsidy per unit of farmland with crop *i*
> $Y_i$: Maximum subsidized units of farmland per crop *i*
> $L^k$: Available units of farmland of farmer *k*

**Variables**

> $x_i^k$: Units of farmland of non-subsidized crop *i* of farmer *k*
> $X_i^k$: Units of farmland of subsidized crop *i* of farmer *k*
> $u^k$: First dual variable of LLM
> $v_i^k$: Second dual variable of LLM.

Finally, adding Equations (12)–(17) as constraints to the ULM (Equations (10) and (11)) formulates the unified bilevel model between governmental subsidies and the agricultural system.

### 2.4. Case Study

The model developed and presented in this article was also applied to the case study of Thessaly, Greece, optimizing the regional agricultural system through subsidy policy in order to minimize energy expenditure of the overall system. According to a 2018 study [19], intense agricultural activity in the area has depleted the aquifer level of the region, which is predicted to only worsen with further climate change expected. Due to the Water-Energy-Food-Climate Nexus [20], reducing the overall energy expenditure, which is in large part attributed to water pumping for irrigation, will reduce both the water expenditure and greenhouse-gas emissions [4]. Data for this case study were extracted from the Hellenic Statistical Authority as well as publications [5,21]. All data refers to 2010. All data used is presented in Table 1, where crops annotated with (I) are irrigated and crops annotated with (NI) are not irrigated.

**Table 1.** 2010 Data for Crops of Thessaly, Greece.

| Crops | Total Farmland (m$^2$) | Yearly Production (kg) | Total Product Value (€) | Crop Yield per Farmland (kg/m$^2$) | Value per Kilo of Crop (€/kg) | Crop Value per Farmland (€/m$^2$) | Water Use (m$^3$/m$^2$) | Energy Use (kJ/m$^2$) |
|---|---|---|---|---|---|---|---|---|
| Rice | 1,823,146 | 1,511,072 | 453,321.55 | 0.829 | 0.3 | 0.25 | 1.134 | 2866.7 |
| Corn | 224,804,489 | 264,718,258 | 47,649,286 | 1.178 | 0.18 | 0.212 | 0.547 | 1851 |
| Other Cereals (I) | 42,558,786 | 335,204,399 | 2,007,327 | 0.635 | 0.2 | 0.127 | 0.5 | 1769.6 |
| Vegetables (I) | 70,909,286 | 248,468,992 | 268,163,519 | 3.227 | 0.8 | 2.58 | 0.658 | 2042.9 |
| Fruit (I) | 112,216,852 | 717,649 | 223,622,093 | 2.214 | 0.9 | 1.99 | 0.7 | 2115.82 |
| Citrus (I) | 755,589 | 25,201,356 | 215,294 | 1.45 | 0.3 | 0.435 | 0.755 | 2211.84 |
| Olives (I) | 129,807,752 | 22,334,289 | 58,467,144 | 0.194 | 2.32 | 0.45 | 0.409 | 1611.81 |
| Potatoes | 10,940,576 | 2,816,550 | 7,817,001 | 2.04 | 0.35 | 0.71 | 0.4 | 1596.43 |
| Lentils | 24,299,555 | 49,097,293 | 4,506,479 | 0.116 | 1.6 | 0.18 | 0.31 | 1440.06 |
| Sugar beets | 7,115,935 | 247,865,051 | 1,472,918 | 6.9 | 0.03 | 0.2 | 0.551 | 1857.56 |
| Cotton (I) | 915,527,563 | 3,755,551 | 148,719,030 | 0.27 | 0.6 | 0.16 | 0.613 | 1965.63 |
| Tobacco | 10,628,109 | 425,891,847 | 13,144,430 | 0.353 | 3.5 | 1.24 | 0 | 903 |
| Wheat | 1,358,529,260 | 67,321,529 | 51,107,021 | 0.313 | 0.12 | 0.03 | 0 | 903 |
| Other Cereals (NI) | 88,896,717 | 10,036,637 | 13,464,305 | 0.312 | 0.2 | 0.06 | 0 | 903 |
| Vegetables (NI) | 215,531,784.3 | 2,874,543 | 2,299,634 | 1.13 | 0.8 | 0.9 | 0 | 903 |
| Fruit (NI) | 2,552,328.26 | 148,621,518 | 133,759,366 | 0.67 | 0.9 | 0.6 | 0 | 903 |
| Citrus (NI) | 278,591.23 | 264,602 | 7,938,073,442 | 0.95 | 0.3 | 0.28 | 0 | 903 |
| Olives (NI) | 218,642,706.2 | 42,448,100 | 98,479,594 | 0.194 | 2.32 | 0.45 | 0 | 903 |
| Nuts | 39,950,885.66 | 13,588,696 | 23,100,784 | 0.34 | 1.7 | 0.58 | 0.51 | 1786.32 |
| Cotton (NI) | 1,605,045.55 | 327,213 | 196,328 | 0.2 | 0.6 | 0.12 | 0 | 903 |

## 3. Results

In Equations (19)–(28) the unified model is formulated and presented.

$$min \sum_{i \in S, k \in K} e_i \left( x_i^k + X_i^k \right) \tag{18}$$

$$s.t \sum_{i \in S, k \in K} s_i X_i^k \leq B \tag{19}$$

$$\sum_{i \in S} \left( x_i^k + X_i^k \right) \leq L^k, \ \forall k \in K \tag{20}$$

$$X_i^k \leq Y_i, \ \forall i \in S, \forall k \in K \tag{21}$$

$$u^k \geq (p_i - c_i)o_i, \forall i \in S, \forall k \in K \tag{22}$$

$$v_i^k + u^k \geq s_i + (p_i - c_i)o_i, , \forall i \in S, \forall k \in K \tag{23}$$

$$\left( \sum_{\iota \in S} \left( x_i^k + X_i^k \right) - L^k \right) * u^k = 0 , \forall k \in K \tag{24}$$

$$(X_i^k - Y_i) * v_i^k = 0, \forall i \in S, \forall k \in K \tag{25}$$

$$\left( u^k - (p_i - c_i)o_i \right) * x_i^k = 0, \forall i \in S, \forall k \in K \tag{26}$$

$$\left( v_i^k + u^k - s_i - (p_i - c_i)o_i \right) * X_i^k = 0, \forall i \in S, \forall k \in K. \tag{27}$$

where:

**Sets**

*S:* Set of available crops $i$
*K:* Set of all farmers $k$

**Parameters**

$p_i$: Profit of crop $i$ per unit of product
$c_i$: Cost of crop $i$ per unit of product
$o_i$: Yield of units of product of crop $i$ per unit of farmland
$s_i$: Subsidy per unit of farmland with crop $i$
$L^k$: Available units of farmland of farmer $k$
$B$: Total available budget for subsidies

**Variables**

$x_i^k$: Units of farmland of non-subsidized crop $i$ of farmer $k$
$X_i^k$: Units of farmland of subsidized crop $i$ of farmer $k$
$u^k$: First dual variable of LLM
$v_i^k$: Second dual variable of LLM
$Y_i$: Maximum subsidized units of farmland per crop $i$.

As shown in Equations (18)–(27), the subsidy value $s_i$ is treated as a parameter and not a variable. To achieve this, the subsidy value is set as the break-even point where the subsidized crop is more profitable than the most profitable non-subsidized crop. This relationship between subsidy value $s_i$ and most profitable non-subsidized crop is shown in Equations (28) and (29).

$$(p_i - c_i)o_i + s_i = A + m , \forall i \in S - \{I\} \tag{28}$$

$$(p_I - c_I)o_I = \max \left[ (p_i - c_i)o_i \right], \forall i \in S = A \tag{29}$$

where:

*I:* Most profitable non-subsidized crop in set $S$.
*m:* A small amount

As shown in Equations (29) and (30), with this subsidy policy utilizing a subsidized crop as a farmer will yield a profit of m over the most profitable non-subsidized crop, ensuring it is always more profitable to take advantage of available subsidies.

## 4. Discussion

In order to apply the model to the case study, certain modifications had to be made to the baseline model presented in order to accommodate available data. First off, an additional constraint was added, signifying a minimum production of each crop, so as to maintain a baseline crop production in the region. The constraint appears in Equation (30):

$$\sum_{k \in K} o_i \left( x_i^k + X_i^k \right) \geq P_{mini}, \forall i \in S. \tag{30}$$

where $P_{mini}$ is the minimum production of crop $i$.

Additionally, since there was insufficient data for each individual farmer, farmers were separated into 5 categories, according to available farmland, and each category was treated as an individual farmer. For this modification, Equations (18)–(20) and (28) were multiplied by parameter $a_k$, which denotes the number of individual farmers that belong in each category $k$. Finally, parameter $e_i$ of Equation (18) is the energy expenditure of crop $i$.

Our analysis shows that government subsidy policies, when scientifically and systemically designed, considering all actors and elements of the system, can indeed determine optimal agricultural production and can successfully direct the individual farmer to crop selection and farmland allocation that is more sustainable, taking into account local resource availability. Other than systemic and inclusive subsidies, subsidies that are targeted can be used to direct farmers to specific behaviors and crop choices that will maximize their profit on the one hand, while also minimizing their associated environmental impact. At the same time, as shown specifically in Equation (30), the model allows the user to add constraints that adapt the model to a specific case study's need; for Thessaly, the constraint that was introduced could support food security and work towards satisfying SDG #2 by maintaining a baseline national production, while also satisfying the aforementioned criteria. It should be noted that, in this article, a proof-of-concept is presented, and in order to achieve reliable real-world results, both a stochastic analysis of the parameters and an extensive historical agricultural dataset of the region would be required.

The model was constructed in the software GAMS IDE (General Algebraic Modeling System Integrated Development Environment) [22] and was solved using the Branch and Bound method [23] for nonlinear Equations (22)–(25). The budget used was 500,000,000 €. Solving the model showed an overall reduction in the energy expenditure of 339 terajoule or 94.17 GWh yearly. Examining the marginal values of Equation (19) showed a reduction of energy expenditure by 677.59 kJ per additional € spent in subsidies, while the most energy inefficient crop was found to be lentils. This application demonstrates the flexibility of the model, since it can be easily modified to adapt to the requirements and data availability of each case study, while also being able to optimize for any required environmental metric, since parameter $e_i$ can be substituted for any metric required without affecting the model's useability.

## 5. Conclusions

In this article, we develop and present a baseline BLP model to describe and minimize the environmental impact of the agricultural sector and its interlinkage with government subsidy policy. We specifically explored the effects of policies and subsidies implemented at the upper-government level on the lower individual farmer level and the mutual interactions. This way we assess the issue of scale in subsidy policies; essentially, we quantify how government-level decisions, such as subsidies, affect the farmer, and how farmer-level decisions, such as crop selection, might affect policy implementation with the goal of minimizing the environmental impact of the overall system. We provide a unified baseline BLP model depicting the two levels and their interaction as well as an application of the model on the regional-scale case study of Thessaly, Greece. The result was a minimization of the energy consumption of the agricultural system, a large part of which was associated with water pumping for irrigation via a Water-Energy-Food-Climate Nexus analysis. The advantage of this specific model is that it does not rely on statistical analysis such as linear regression or on heuristic algorithms but presents a stricter optimization approach to an important issue through a vector that is not well explored. To expand this work, an economic analysis can be implemented to also model the changes in crop prices with different crop productions, further modifying the objective functions and constraints and produce more accurate results. A multi-objective optimization approach can also be considered in order to minimize many different environmental-impact parameters simultaneously while also considering their Nexus interlinkages. Finally, the temporal structure of the model can be altered to be monthly, in order to enable the "flattening" of environmental-impact



parameters throughout the year; this would, for example, enable us to consider each crop's water usage "spikes" in specific months and produce a more uniform consumption and impact assessment, where it is relevant.

**Author Contributions:** K.Z.: Conceptualization, Methodology, Software, Data Curation, Validation, Formal Analysis, Investigation, Resources, Visualization, Writing—Original Draft; K.P.: Writing—Review & Editing, Supervision. All authors have read and agreed to the published version of the manuscript.

**Funding:** The work described in this paper has been conducted within the project NEXOGENESIS. This project has received funding from the European Union's Horizon 2020 research and innovation programme under Grant Agreement No. 1010003881 NEXOGENESIS. This paper and the content included in it do not represent the opinion of the European Union, and the European Union is not responsible for any use that might be made of its content.

**Informed Consent Statement:** Not applicable.

**Data Availability Statement:** The dataset used in this article is at: https://data.mendeley.com/datasets/9x7wn24rrp/1, accessed on 7 July 2021.

**Conflicts of Interest:** The authors declare no conflict of interest.

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
