# Peer review of "A Bilevel Linear Programming Model for Developing a Subsidy Policy to Minimize the Environmental Impact of the Agricultural Sector"

_sustainability, doi:10.3390/su14137651_

Round 1

Reviewer 1 Report

Abstract

  • Your abstract should clearly state the essence of the problem you are addressing, what you did and what you found and recommend. That will help a prospective reader of the abstract to decide if they wish to read the entire article.

Introduction

  • What is the lack in the current literature and what kind of gap does this research intend to close?
  • An updated and complete literature review should be conducted. The challenges and barriers in the technological aspect should be briefly addressed to highlight the novelty and significant contribution of this study.

Results

  • Results are well presented however discussion part of this part of the manuscript miss some important aspects. Only a few papers are cited and used here. If possible, I would suggest changing the Result and Discussion by showing a wider view of the obtained result in the context of literature or giving an explanation on the limitation of a small number of literature sources. In my opinion, this part of the manuscript would benefit from adding more other studies.

Conclusion

  • Conclusion can go deeper; it would be more interesting if the authors focus more on the significance of their findings
  • The suggestion for future research is missing in the conclusion section.

Author Response

Please see attached file for our response. Thank you for your comments.

Reviewer 2 Report

This MS introduced a bilevel linear programming (BLP) model for developing a subsidy policy for the agricultural sector. It is interesting. However, it still need major revision to get publication. Firstly, this research tried to minimize the environmental impact in a sets of cropping systems. Here, only water and energy have been included in the environmental indexes. But the significant environmental effects of green house gases emission, soil erosion, eutrophication, etc, did not been concerned. So, it is not enough to address the environmental impact. Second, where has been significantly verified in the model? It looks same as the method introduction. Thirdly, the results and discussion can be put together. Fourthly, what is the main case results? It is difficult to find advantages to use the current model.

Author Response

(The authors gave the same response as above.)

Reviewer 3 Report

sustainability-1717010-peer-review-v1
A bilevel linear programming model for developing a subsidy policy to minimize the environmental impact of the agricultural sector.

In this paper, author shares a methodology based on bi-level linear programming for developing a subsidy policy to minimize the environmental impact of the agricultural sector. The topic is interesting. However, the paper needs some corrections to meet expectations of a journal. Some of my concerns are as follows:

  1. The benefits of the proposed method have been demonstrated clearly. What’s the limitation of the method? Are there other ways that the results can be further improved? One or two remarks should be given to discuss it in detail.
  2. The study-organizing paragraph is missing. Author(s) should include a 'study organizing paragraph' in the end of the introduction section. The paragraph helps reader to understand the overall sequence and flow of your paper.
  3. The "Where" below Eq. (6) should be "where". Remove the similar problems in your paper.
  4. There is no comparison with new model and some existing models. Comparative analysis needs to be explained.
  5. Literature review, in my point of view is weak, which required to improve and strengthen. Author(s) need to cite more latest researches in the relevant field to provide an up-to-date picture of work. Following articles can be cited in introduction and literature review sections to enrich this parts:
  6. The subject might be interesting for some researchers but the authors do not motivate the subject by explaining its relevance compared to the literature. There are many papers that are cited in the introduction but they are rather given as a list of papers without any further explanation or motivation. For a stronger submission, I recommend the authors to write a stronger introduction section where, using the relevant literature, they emphasize the importance of the subject as well as the contribution of the paper, clearly. 
  7. Finally, in the conclusion section, the main questions should be answered in conclusion section: a) who needs this, b) what is the contribution of your paper, c) what benefit have investors if they decide to use the proposed approach in their portfolio optimization.
  8. I also recommend the authors to professionally get the paper proofread, as I have noticed sentences with typos and inappropriate choice of words. For instance, “the dual constraints and the complementary slackness conditions of the LLM is presented.”

***

Author Response

(The authors gave the same response as above.)

Reviewer 4 Report

In this paper, the authors introduced a bilevel linear programming (BLP) approach to minimize the environmental impact of the agricultural sector, by exploring and quantifying the effects of governmental subsidy policy frameworks on farmers and how decisions such as crop selection of the latter, might affect policy regime. The proposed mathematical approach is applied to the case study of Thessaly where the results indicate a remarkable minimization of energy expenditure for a theoretical subsidy budget. The strength of the baseline BLP model relies on the fact that it is flexible and adjustable to different conditions and targets highlighting its replicability potential at different optimization scales in the government-agriculture interface.

The introduction section is well-organized and structured with an extended literature review on previous research efforts based on BLP modelling, while the authors provide sufficient explanations on how their approach progresses beyond the state-of-the-art by highlighting its potential to address different environmental goals on the Water-Energy-Food-Climate Nexus.

Material and methods section describes in detail the modelling levels, while the results, discussion, and conclusions are well-written, with adequate explanations when needed and a satisfactory overall flow.

General Comments:

- I would suggest a graphical/schematic overview picture of the proposed BLP model including the basic components of the optimization procedure. This is optional and if the authors find it applicable, I recommend placing it in subsection 2.3 “Formulation of Single Level Model”.

- In Table 1 please add units in the second column “total farmland”; probably it’s in m2. Moreover, I would suggest adding commas in the numbers of the table delineating thousands for a smoother reader experience. Lastly, please replace “m3” and “m2” with “m2” and “m3” in the column’s labels of the table.

Line-specific comments:

Line 66: Instead of “and” please write “an”,

Line 126: Please replace “e_i” with “ei”,

Lines 167 & 170: Please replace “s_i” with “si”,

To this end, I recommend this manuscript for publication with minor revisions, as described above.

Author Response

We would like to thank this reviewer for their review. All line specific comments were addressed, as well as a figure denoting the logic behind the multi objective function optimization problem before it is merged.

Round 2

Reviewer 1 Report

Accept for publication without revision.

Author Response

We would like to thank this reviewer for his time and effort.

Reviewer 2 Report

The current MS showed very weak structure. The introduction is too long and did not figure out the main scientific problems. For the method, it showed very long introduction on the model. For the case study, it did not show the main results and the advantages of the revised model. For the results, it would belong to method section. For modeling, it generally need calibration and validation. But ,this MS did not this part. So, we can not judge the model work well or not.

Author Response

We would like to thank the reviewer for his review and comments.

The introduction was elongated due to requests from other reviewers. However, some cuts have been made to make it more concise. Additionally, many syntax and other errors were corrected. 

In line 40 of the introduction, a line has been added to further clarify the purpose of the manuscript. The purpose of the manuscript, and therefore the results section, is a proof of concept baseline model, that merges a multi objective function problem into a single level model that can optimize for any environmental variable necessary. The model is not to be used as-is, but rather adapted to a specific case study's needs and data availability through different constraints and parameter implementation. As such, we feel having the final model in the result section is appropriate, as it is the main focus of the manuscript, not the case study application.

We would like to once again thank the reviewer for his time and effort.

Reviewer 3 Report

Manuscript ID: sustainability-1717010-peer-review-v2

Title: A bilevel linear programming model for developing a subsidy policy to minimize the environmental impact of the agricultural sector.

Dear Editor,

Author shares a study on a bilevel linear programming model for developing a subsidy policy to minimize the environmental impact of the agricultural sector. The topic is interested. However, this is an ordinary manuscript.  There is no strong result and application in this manuscript.  Some comments are the following:

  1. The abstract isn't sufficiently concise and informative.
    2. The purpose of the article doesn't clearly state in the introduction.
    3. The article achieve doesn't declare the purpose.
    4. The article doesn't show clarity of presentation.
    5. The English and syntax of the article are not satisfactory.
    6. The document is not concise.

Thank you for giving me this opportunity.

Regards,

Author Response

We would like to thank the reviewer for his review and comments.

  1. We feel the abstract can not be shortened further without missing important information either about the problem framing, the purpose of the article and its results.
  2. and 3. In line 40 of the introduction a line has been added to make the purpose clearer.

     4. and 5. Many syntax and other small errors were corrected in the manuscript.

    6. The document has been shortened somewhat, however it was originally elongated due to requests from other reviewers, especially the introduction. However, some cuts have been made. 

We would like to once again thank the reviewer for his time and effort.

Round 3

Reviewer 2 Report

The current version is still far to get publication due to the bad structure.

Author Response

According to the editor instructions, we will only reply to the comments of reviewer 3.

Reviewer 3 Report

sustainability-1717010-peer-review-v3

Title: A bilevel linear programming model for developing a subsidy policy to
minimize the environmental impact of the agricultural sector.

The authors have presented a very good work in revised version. However, I have some concerns must be addressed before accepting this manuscript.

Comments

1.      The "Where" below Eq. (9) and Eq. (11) should be "where". Remove the similar problems in your paper.

2.      The text written in Figure 1 are not readable.

3.      Please add a summary paragraph in the last paragraph of Introduction section, by mentioning which section number describes what?

4.      There is no comparison with new model and some existing models. Comparative analysis needs to be explained.

5.      Some typos were found.

***

Author Response

  1. The problems have been identified and addressed in the revised manuscript.
  2. The figure has been altered to improve readability in the revised manuscript.
  3. A summary paragraph has been added in the revised manuscript.
  4. A review of existing models has been done in the literature review of section 1. No further changes have been made.
  5. Typos have been identified and addressed in the revised manuscript.